# Low-Speed Clinorotation of *Brachypodium distachyon* and *Arabidopsis thaliana* Seedlings Triggers Root Tip Curvatures That Are Reminiscent of Gravitropism

**DOI:** 10.3390/ijms24021540

**Published:** 2023-01-12

**Authors:** Shih-Heng Su, Alexander Moen, Rien M. Groskopf, Katherine L. Baldwin, Brian Vesperman, Patrick H. Masson

**Affiliations:** 1Laboratory of Genetics, University of Wisconsin-Madison, 425 Henry Mall, Madison, WI 53706, USA; 2Kate Baldwin LLC, Analytical Design, Cross Plains, WI 53528, USA

**Keywords:** clinostat, gravitropism, gravisensitivity, *Arabidopsis thaliana*, *Brachypodium distachyon*, presentation time, perception time, realization time, clinostat effect

## Abstract

Clinostats are instruments that continuously rotate biological specimens along an axis, thereby averaging their orientation relative to gravity over time. Our previous experiments indicated that low-speed clinorotation may itself trigger directional root tip curvature. In this project, we have investigated the root curvature response to low-speed clinorotation using *Arabidopsis thaliana* and *Brachypodium distachyon* seedlings as models. We show that low-speed clinorotation triggers root tip curvature in which direction is dictated by gravitropism during the first half-turn of clinorotation. We also show that the angle of root tip curvature is modulated by the speed of clinorotation. *Arabidopsis* mutations affecting gravity susception (*pgm*) or gravity signal transduction (*arg1*, *toc132*) are shown to affect the root tip curvature response to low-speed clinorotation. Furthermore, low-speed vertical clinorotation triggers relocalization of the PIN3 auxin efflux facilitator to the lateral membrane of *Arabidopsis* root cap statocytes, and creates a lateral gradient of auxin across the root tip. Together, these observations support a role for gravitropism in modulating root curvature responses to clinorotation. Interestingly, distinct *Brachypodium distachyon* accessions display different abilities to develop root tip curvature responses to low-speed vertical clinorotation, suggesting the possibility of using genome-wide association studies to further investigate this process.

## 1. Introduction

Most plants use gravitropism to guide the growth of their organs, directing primary roots vertically downward into the soil, primary shoots upward above ground, and secondary organs laterally, at an angle from the vertical named the gravity set point angle (GSA), to effectively explore their immediate surroundings. Overall, this process contributes to a plant’s ability to anchor itself into the soil, actively seek water and nutrients within its growth substrate, exchange gases, photosynthesize, and reproduce. Consequently, gravitropism is a key environmental response that allows plants to thrive under diverse environmental conditions, thereby contributing to plant fitness and productivity in their natural ecological niches as well as in agricultural settings.

Gravitropism is composed of several successive phases that include gravity susception, believed to involve the sedimentation of heavy starch-filled plastids (amyloplasts, also named statoliths) in specialized cells (the statocytes) located in the starch sheath of shoots and in the columella cells of the root cap. This is followed by transduction into a biochemical signal through a process that remains largely unexplained, the formation of an auxin gradient across the gravistimulated organ, transmission of the gradient to the site of response, and differential cell elongation on opposite flanks of the organ, leading to a curvature response, usually upward for shoots and downward for roots (reviewed in [1]).

The molecular mechanisms underlying the initial phases of gravity sensing and signal transduction in the statocytes remain largely unknown. However, this process leads to a fast cytoplasmic alkalinization of the statocytes upon gravistimulation, accompanied by a change in polarity of the cell associated with the transcytotic relocalization of the auxin efflux facilitators PIN3 and PIN7 to the new bottom membrane [2,3,4,5]. This relocalization is dependent upon the proper functioning of key regulators in the statocytes, including ALTERED RESPONSE TO GRAVITY 1 (ARG1) and its paralog ARG1-like 2 (ARL2) [6,7], MODIFIER OF *arg1* 1 (MAR1/TOC75) and MAR2/TOC132 [8], LAZY, RCC1-LIKE DOMAIN (RLD) [9,10,11], and ADENOSINE KINASE 1 (ADK1) [12]. Additionally, mutations in genes that encode enzymes involved in starch biosynthesis, such as *pgm-1*, affect gravitropism mostly by decreasing the density of statocyte amyloplasts, thereby affecting gravity susception [13]. Finally, PIN protein phosphorylation at conserved Ser residues has been shown to also modulate PIN3 relocalization in the statocytes and its auxin transport activity (reviewed in [14]).

In roots, PIN3 and PIN7 relocalization in the upper layers of columella cells in the root cap upon plant reorientation within the gravity field allows the establishment of a lateral auxin gradient across the cap, which is transported through the lateral root cap and epidermal cells to the transition zone, where it promotes differential cellular elongation between the top and bottom flanks, leading to downward curvature. Rootward auxin transport in the peripheral tissues of the root is mediated by AUX1 and PIN2 auxin influx and efflux facilitators, respectively, as well as ABC transporters [15,16,17,18]. The PIN2 protein is localized in the plasma membrane mostly on the shootward side of the lateral cap and epidermal cells, dictating the shootward transport of auxin through cell files. Its recycling between the endosome and plasma membrane, and the regulation of its stability, are critical for its function in auxin transport and gravitropism [19]. Its abundance within the plasma membrane of transporting cells is enhanced by increased auxin levels and GOLVEN peptide signaling on the bottom flank, a process that reinforces the auxin gradient across the root tip as it moves toward the transition zone [19,20]. Furthermore, PIN2-mediated reflux of auxin from the peripheral tissues back into the vasculature stream toward the root tip also occurs at the transition zone, contributing significantly to the modulation of root gravitropism [21].

The formation of an auxin gradient across the root tip and its transport to the transition zone and main segment of the elongation zone can be followed using live reporters. One of these reporters, named *DII-VENUS*, encodes a protein that carries the cis-acting motifs needed for recognition by the SCF^TIR1^ ubiquitin–ligase complex in the presence of auxin, leading to its degradation by the proteasome [22]. Hence, the cellular fluorescence signal generated by the *DII-VENUS* reporter rapidly decreases as a consequence of auxin-induced reporter degradation by the proteasome [22,23].

A better understanding of the key molecular mechanisms that modulate gravity sensing and signal transduction in plants requires an accurate method to evaluate organ gravisensitivity, such that genotypes with altered gravisensitivity can be identified and further characterized. Early studies suggested that the curvature response of a plant organ to gravistimulation varies linearly as a function of the logarithm of the dose of gravistimulation, itself defined as the product of gravity intensity by time of stimulation (g.s) [24]. Quantifying tip angle for distinct doses of initial gravistimulation led to dose–response curves that fit logarithmic functions. Extension of the corresponding best-fit functions to the time axis under 1 g allowed the determination of the **presentation time**, defined as the threshold stimulation time needed to produce curvature (reviewed in [25]). This method has been used extensively to estimate the gravisensitivity of plant organs despite ignoring the fact that very short periods of gravistimulation well below the presentation time can still induce gravicurvature when applied repeatedly. The threshold time of intermittent gravistimulation leading to gravicurvature (**the perception time**) is shorter than 1 s, suggesting the gravisensing machinery is exquisitely sensitive to reorientation [26,27,28].

A re-evaluation of some of the data presented in previous presentation time studies revealed that a hyperbolic model better fits the observed data than the initially used logarithmic model [25]. Such hyperbolic models intersect the time axis at the origin, suggesting the slope of the best-fit hyperbolic model at the origin might provide a better representation of gravisensitivity than the presentation time [25]. Interestingly, a recent study using a centrifugation approach coupled with growth kinematics imaging provided support for a contribution of **organ inclination**, rather than gravity intensity, in shoot gravity sensing [29].

The presentation and perception times described above are much shorter than the time needed for initiation of the resulting curvature. This implies that the initial gravistimulation should be followed by a period without stimulation to allow the plant organ to develop curvature that is commensurate with the initial stimulus. The microgravity conditions encountered during spaceflight or at the International Space Station (ISS) are ideal for the implementation of such experiments [30,31,32]. Unfortunately, using either platform is very expensive and access to them is very limited. Furthermore, the use of centrifuges to create fractional gravity in space implies the creation of sharp gravity gradients between the center of the centrifuge and its periphery, which can have an unwanted impact on plant responses.

To overcome these difficulties, researchers have used low-speed clinostats to mimic microgravity during the curving phase of the graviresponse. Indeed, these continuously rotating devices average the plant orientation relative to the gravity vector during each rotation cycle (reviewed in [33]). However, using this approach to define organ gravisensitivity is heavily reliant on the assumption that clinorotation does not have an impact of its own on root growth behavior. Unfortunately, published data may suggest otherwise.

First, it is important to note that research teams have used clinostats to mimic microgravity in a variety of different ways, including rotating the plants at a range of angular velocities (from 0.33 to 60 RPM), with one rotation axis parallel or perpendicular to the longitudinal growth axis of the plants (horizontal vs. vertical clinorotation, respectively), or rotating in space (three-dimensional clinostats or random positioning machines) (reviewed in [34]). A recent and careful investigation of the impact of various forms of clinorotation on plant organs growth revealed that slow horizontal clinorotation at 1 RPM induces stress responses in root statocytes and lower endodermal cells, whereas fast clinorotation in the range of 60 RPM leads to the directional root growth response to a force that results from both gravity and the centrifugal acceleration [34]. These results suggest a need to re-evaluate the use of clinostats as microgravity mimics in experiments aimed at quantifying plant gravisensitivity.

Considering that previous experiments aimed at evaluating the sensitivity of plant organs to gravistimulation used low-speed clinorotation to allow the development of a proportional curvature response to defined doses of gravistimulation applied before clinorotation, and given that presentation times of 0.4–5 min were previously reported for *Arabidopsis thaliana* organs using these methods [12,13,35,36], we set up an experiment to investigate whether low-speed clinorotation might itself trigger a directional root growth response without an initial dose of gravistimulation before clinorotation. Indeed, if seedlings are positioned vertically on a clinostat and rotation is initiated at a speed of 0.5 RPM, their organs will be exposed to a unidirectional gravistimulus over the first half-turn of clinorotation, which should last 60 s, before being exposed to a gravistimulus from the opposite direction. If the first half-turn of gravistimulation is sufficiently long enough to induce a persistent polarity in the statocytes, it should be sufficient to promote curvature at the end of clinorotation. In fact, our previous experiments aimed at evaluating the effect of clinorotation on *Arabidopsis* root skewing on hard agar surfaces uncovered the existence of clinostat-dependent root tip curvatures that were compatible with this “clinostat effect” [37].

To better understand the effects exerted by low-speed clinorotation on root growth behavior and their connections with gravitropism, we exposed *Arabidopsis thaliana* (a dicot model) and *Brachypodium distachyon* (a genetic model for monocot plants) seedlings, growing on agar-solidified medium, to low-speed clinorotation in the absence of a previous gravistimulus (defined as seedling reorientation away from the vertical), and analyzed their root tip responses to such clinorotation. We demonstrate the development of root tip curvature in both species at low speeds of clinorotation, and show that mutations affecting early steps of gravity sensing and/or signal transduction interfere with this clinostat effect. We also show that low-speed clinorotation leading to root tip curvature promotes the relocalization of the PIN3 auxin efflux facilitator to the side membrane of the root cap statocytes and the development of a lateral gradient of auxin across the root tip. We conclude that gravitropism contributes to the clinostat effect on root growth direction under low-speed clinorotation, and discuss the potential implications of these findings for our understanding of root gravisensitivity.

## 2. Results

### 2.1. Low-Speed Clinorotation Triggers Directional Root Tip Curvature in Arabidopsis thaliana and Brachypodium distachyon Seedlings

To investigate the effect of clinorotation on root growth behavior without prior gravistimulation, we subjected vertically grown *Arabidopsis thaliana* and *Brachypodium distachyon* seedlings to low-speed clinorotation in darkness, and then measured root tip reorientation at the end of the treatment. For *Arabidopsis thaliana*, we used seedlings from the Col and Ws ecotypes because several mutants with defects in the early steps of gravity sensing and signal transduction are available in these backgrounds (see below).

In the first assay, *Arabidopsis thaliana* seedlings growing on the surface of an agar-based medium in Petri dishes were subjected to low-speed (0.2 RPM) vertical clinorotation in clockwise or counterclockwise directions. The results summarized in Figure 1A,B (upper halves of the panels) indicate that the roots of Col seedlings exposed to clockwise clinorotation curved to the right, eventually reaching tip angles of around 25.5 ± 4.9 degrees, whereas those exposed to counterclockwise clinorotation curved to the left, reaching tip angles of −38.5 ± 7.1 degrees. A similar effect was also observed when Ws seedlings were tested in this assay (Figure 2A,B). The roots of seedlings that were maintained in the clinostat chamber without being clinorotated (vertical controls, labeled ST in Figure 2) did not develop tip curvature, suggesting that clinorotation is responsible for root reorientation. Clinorotation did not significantly affect the growth of wild type Col and Ws roots (Appendix A).

To determine whether this effect is *Arabidopsis*-specific, we subjected *Brachypodium distachyon* seedlings (Bd21-3 accession) to a similar experimental protocol. Results shown in Figure 1A,B (lower halves of the panels) indicate that the seedling roots of this monocot model also responded to clockwise clinorotation by curving rightward (tip angles of 12.5 ± 2.9 degrees), whereas those exposed to counterclockwise clinorotation curved to the left (eventually reaching tip angles of −14.8 ± 1.4). The roots of stationary seedlings did not develop tip curvature (Figure 1A,B). Clinorotation did not significantly affect *Brachypodium* root growth relative to the stationary growth control (*t*-test *p*-value = 0.41 and 0.22). This result demonstrates that low-speed clinorotation also promotes root tip curvature in *Brachypodium distachyon*.

A second experimental setup was designed to confirm the existence of a clinostat effect on root growth direction, relying on the two-chamber system described in the Materials and Methods section and illustrated in Appendix A. In this system, the plates loaded in chamber 1 were oriented such that seedlings were in an upright position, whereas those loaded in chamber 2 were upside down. During the first half-turn of clinorotation, the right and left sides of seedlings located in chamber 1 and chamber 2, respectively, were directed toward the gravity vector (Appendix A). When clinorotated clockwise at 0.2 RPM, chamber 1 seedlings curved to the right, whereas those exposed to counterclockwise clinorotation curved leftward (tip angles of 22.6 ± 5.1 and −25.9 ± 9.4 degrees, respectively; Appendix A). The opposite results were observed in chamber 2, where seedlings exposed to clockwise rotation curved leftward and those exposed to counterclockwise rotation curved rightward (tip angles of −19.2 ± 11.3 and 14.2 ± 8.6 degrees, respectively; Appendix A).

To further confirm these observations, we subjected Bd21-3 *Brachypodium distachyon* seedlings to 0.2 RPM **parallel clinorotation**, where plates were oriented parallel to the clinostat axis, with seedlings oriented vertically at the beginning of the experiment (Appendix A). Plates were loaded back-to-back into the chamber, and clinorotation was immediately initiated. Half of the plates had seedlings with their air-exposed side facing the gravity vector during the first half-turn of clinorotation, whereas the other half had seedlings with their medium-facing side directed toward gravity. At the end of clinorotation, the root tips of a majority of tested seedlings in the first group (five out of seven) had curved away from the surface (Appendix A, left), whereas those from the second group (with medium-facing side directed toward gravity during the first half-turn of rotation) tended to either penetrate the agar-based medium (five out of seven seedlings tested) or skew on its surface.

We also subjected Bd21-3 seedlings to 0.2 RPM **horizontal clinorotation**, where the plates were positioned vertically in the clinostat chamber with seedlings oriented horizontally, parallel to the clinostat axis (Appendix A). Here again, plates were loaded back-to-back into the chamber, and clinorotation was immediately initiated. At the end of clinorotation, the root tips of a vast majority of the tested seedlings (15 out of 16) whose air-exposed side faced the gravity vector during the first half-turn of clinorotation had curved away from the surface (Appendix A, left), whereas those with medium-exposed surfaces directed toward gravity during the first half-turn tended to penetrate the agar-based medium (8 out of 8 seedlings tested) (Appendix A, right).

Taken together, the results from these experiments demonstrate that low-speed clinorotation triggers directional root tip curvature dependent upon both the orientation of the seedlings at the beginning of clinorotation and the polarity of rotation (clockwise or counterclockwise, Figure 1; Appendix A). In all these experiments, the roots curved in a direction defined by the side that faced gravity during the **first half-turn of clinorotation**, starting from a vertical root, suggesting a role for gravitropism in this response.

### 2.2. Mutations Affecting Early Steps of Gravity Signal Transduction Interfere with the Clinostat Effect

To test the contribution of gravitropism to the clinostat effect, we evaluated root curvature responses to clockwise vertical clinorotation of *Arabidopsis thaliana* mutants with defects in early phases of gravity sensing and/or signal transduction, including two *arg1* mutant lines (*arg1-2* and *arg1-3*; [7]) and the corresponding transgenic rescue line (*arg1-2[35S:ARG1]*), three *toc132* mutant lines (*toc132-3*, *toc132-4,* and *mar2-1*; [8]), and the starch-deficient *pgm-1* mutant [13]. While the roots of control wild type seedlings (Col and Ws) displayed strong curvature responses to clinorotation at speeds of 0.2 RPM relative to stable non-clinorotated controls (ST), those of *pgm*, *arg1*, and *toc132* mutant seedlings did not display much of a clinostat effect (Figure 2; Appendix A; Table 1). Their average root tip curvature upon clinorotation was significantly different from that of the wild type (*t*-test *p*-value < 0.05; Table 1). The root tips of *pgm* and *arg1* mutant seedlings also displayed a more random orientation than the wild type under control growth conditions (no clinorotation; ST), indicative of gravitropism defects, whereas those of *toc132*/*mar2* seedlings did not, in agreement with a previous report [8] (F-test *p*-value < 0.05; Table 1; Figure 2; Appendix A). The roots of transgenic homozygous *arg1-2* mutant seedlings expressing the *35Sp:ARG1* transgene displayed a significant clinostat effect (Table 1), demonstrating phenotype rescue by the transgene (Figure 2). Clinorotation did not alter root growth in either wild type or mutant seedlings relative to stationary controls (Appendix A). Taken together, these results support a contribution of gravitropism to the clinostat effect.

### 2.3. The Speed of Clinorotation Impacts the Clinostat Effect

If gravitropism is responsible for the clinostat effect, the initial period of exposure to directional gravistimulation, hence the speed of clinorotation, should impact the level of root tip curvature at the end of the experiment, with shorter periods of initial exposure (associated with higher speeds of clinorotation) resulting in lower curvatures. To test the validity of this assumption, we subjected wild type Col seedlings to clinorotation at speeds of 0.1, 0.2, and 0.5 RPM, and evaluated the resulting root tip curvatures at the end of clinorotation. In Figure 3, panels A and B show that lower clinorotation speeds lead to increased root tip curvature angles (more significant clinostat effects).

To better understand the relationship between rotation speed and resulting root tip curvature, we plotted the average values of root tip angle (Figure 3C) and total tip curvature (Figure 3D) (*Y* axis) over the duration of the first half-turn of clinorotation (*X* axis), and then modeled the results with a best-fit logarithmic function (a method analogous to that used to estimate the presentation time [25]). The best-fit models identified in these experiments closely matched the observed data, with R^2^ values often exceeding 0.9 (Figure 3). Their intersection with the *X* axis defines the minimal period of half-turn clinorotation needed to determine the polarity of root tip curvature. We named this parameter the **realization time** (T_R_). In the experiments shown in Figure 3C, the roots of wild type Col seedlings displayed a T_R_ of 35 s. Two additional independent trials led to similar T_R_ estimates for *Arabidopsis* wild type Col roots (Appendix A). On the other hand, similar experiments carried out with two distinct *mar2* (*toc132*) mutants showed longer realization times of 54.7 s and 62.6 s, respectively (Appendix A). No T_R_ values could be assigned to the *pgm1* and *arg1* mutants because both developed only minor, insignificant curvature responses to low-speed clinorotation that did not fit with a logarithmic model (Appendix A).

One limitation of the experiment described in Figure 3 is that it attempts to correlate the stimulus provided during the first half-turn of clinorotation with the final angle of curvature even though the same stimulus is provided incrementally at each cycle of clinorotation throughout the whole experiment. To investigate the possible contribution of additional rounds of clinorotation (beyond the first half-turn) to curvature specification, we modified the clinorotation experiment to first include one or more curvature-inducing cycles at the beginning of the experiment (a phase we call “inductive”), and followed it with a phase of non-inductive 3 RPM clinorotation to allow curvature development (schematized in Figure 4A, left panel). The inductive phase included one, two, three, four, or five cycles of clinorotation at a speed of 0.2 RPM. Two controls included one experiment without an inductive phase (all cycles at 3 RPM), and a second experiment with all cycles being inductive at 0.2 RPM. The results shown in Figure 4 indicate that continuous clinorotation at 3 RPM is not sufficient to induce a significant root tip curvature relative to non-clinorotated controls (ST), as expected. Similarly, neither one, two, or three cycles of inductive clinorotation were insufficient to promote the clinostat effect. However, an inductive phase of four or five cycles at 0.2 RPM followed by 18 h at 3 RPM led to root tip curvatures similar to those generated by continuous clinorotation at 0.2 RPM (Figure 4). Hence, only four inductive clinorotation cycles at 0.2 RPM are sufficient to trigger strong root tip curvature similar to that triggered by continuous rotation at this speed (Figure 4).

### 2.4. Low-Speed Clinorotation Triggers the Formation of an Asymmetric Auxin Gradient across the Root Tip

If the clinostat effect is a consequence of gravitropism, low-speed clinorotation should trigger an asymmetric auxin redistribution across the root tip, preceding the initiation of root tip curvature. To investigate this possibility, we clinorotated, at 0.2 RPM, wild type *Arabidopsis thaliana* Col seedlings transformed with the *DII-VENUS* auxin reporter [22].

Analysis of VENUS expression along the root tip of control non-rotated seedlings (ST) revealed symmetrical signals in epidermal and lateral cap cells on opposite flanks of the root tip up to the transition zone (Figure 5A). The symmetrical signal was also found across the root tips of seedlings exposed to non-inductive 3 RPM clinorotation for 1, 2, or 6 h (Figure 5A). As expected, strong signal asymmetry was observed across the root tips of 1 h gravistimulated seedlings, with a dramatic signal decrease on the lower side of the root ([22]; Figure 5A; Appendix A). Interestingly, an asymmetrical VENUS signal was also apparent across the root tip after 1 h of low-speed clinorotation (0.2 RPM), with a lower signal on the root tip side that was directed toward gravity during the first half-turn of clinorotation (Figure 5A; Appendix A). Signal asymmetry persisted after 2 and 6 h of clinorotation, although the asymmetry was weaker at these later time points than after 1 h (Figure 5A, Appendix A). Signal quantification confirmed significant signal asymmetry upon both GS and low-speed (0.2 RPM) clinorotation (Figure 5A, Appendix A). These results are compatible with the development of an auxin gradient across the root tip of low-speed-clinorotated *Arabidopsis* seedlings, with auxin accumulation on the side of the root facing gravity during the first half-turn of clinorotation.

If gravitropism is truly responsible for the clinostat effect, the auxin gradient developed across clinorotated roots should originate from the root cap columella cells, where a redistribution of plasma-membrane-associated PIN3 and PIN7 proteins should occur (reviewed in [1]). To test this assumption, we subjected Col seedlings expressing the PIN3-GFP transgene [38] to the same low-speed clinorotation assay and analyzed GFP fluorescence signal in the root cap at 0, 1, 2, and 6 h following the onset of clinorotation. The results shown in Figure 5B show a mild increase in PIN3-GFP signal intensity in the lateral membrane of tier 2 and 3 columella cells at the side of the roots that faced down during the first half-turn of clinorotation; this occurred within 1 h of clinorotation. The asymmetry of this PIN3-GFP signal was milder than the one observed in response to normal gravistimulation. It was still present after 2 and 6 h of clinorotation, but it was less pronounced after 6 h. No signal asymmetry was found when the roots were not stimulated (ST), or when the roots were clinorotated at the higher speed of 3 RPM, which is non-inductive (Figure 5B).

To evaluate the relative timing of PIN3-GFP relocalization in the root cap statocytes and DII-VENUS gradient formation across the root tip relative to curvature initiation upon clinorotation, we quantified the root tip angle of the same clinorotated PIN3-GFP-expressing seedlings at different times following 0.2 RPM clinorotation onset. The results shown in Appendix A indicate the appearance of a significant curvature after 5 h of clinorotation. Although the kinetics of fluorescent marker changes have to be taken with caution considering the time needed for marker maturation during synthesis and/or marker degradation [22], we note that the appearance of asymmetrical PIN3-GFP and DII-VENUS signals in the root tips precede the initiation of the corresponding curvature (Figure 5A,B; Appendix A). These results are compatible with the few initial rounds of clinorotation leading to PIN relocalization toward one side of the columella cells dictated by the gravistimulus imposed during the first half-turn of clinorotation, lateral auxin transport, and the resulting curvature response.

### 2.5. The Bd21-3 and Ron-2 Brachypodium distachyon Accessions Are Differentially Responsive to Low-Speed Clinorotation

Considering our observations of *Arabidopsis thaliana* mutants with distinct abilities to develop root tip curvatures in response to low-speed clinorotation (Figure 2 and Appendix A), we wondered whether the genetic variation existing between natural populations of *Brachypodium distachyon* might also be associated with distinct abilities to respond to clinorotation. To address this possibility, we subjected seedlings of the *Bd21-3* and *Ron-2* accessions to different speeds of vertical clinorotation (0, 0.1, 0.2, and 0.5 RPM), and then evaluated the resulting root tip curvatures. The results shown in Figure 6A demonstrate strong and weak responses to 0.1 and 0.2 RPM clinorotations, respectively, leading to a realization time (T_R_) of 91 s for *Bd21-3*. Ron-2, on the other hand, displayed no root tip curvature responses to low-speed clinorotation, preventing us from assessing the realization time for this accession. These results demonstrate the existence of a large variability between *Brachypodium* accessions for their abilities to develop root tip curvatures under low-speed clinorotation.

## 3. Discussion

Since their inception by von Sachs in 1879 [39], clinostats have been advertised as useful systems to mimic microgravity on Earth. By constantly rotating around one or two axes, these devices simulate weightlessness by averaging a plant’s orientation relative to gravity at each rotation cycle. Consequently, the plant is never exposed to gravity from one side for long enough to trigger a tropic response. Under these conditions, the plant organs are expected to display growth behaviors that are comparable to those seen under microgravity as long as the speed of rotation remains sufficiently low to keep the centrifugal force below the limit of detection by the organism (reviewed in [34]). In plants, clinostats have been used rather effectively to evaluate the effect of microgravity on processes as diverse as tropic responses, plant morphology, metabolism, organ growth, cell division, organelle morphology, and nutrient uptake, to cite just a few examples [25,40,41,42]. They have also been extensively used to investigate plant organs’ sensitivity to gravistimulation (reviewed in [25]).

Considering that repeated short periods of gravistimulation can trigger a gravitropic response [25,26,27], we wanted to reassess the assumption that low-speed clinorotation truly simulates weightlessness. Indeed, if a vertical plant is suddenly exposed to low-speed vertical clinorotation, its organs will be gravistimulated from the same side over a period corresponding to the first half-turn of rotation. If this initial half-turn of clinorotation is sufficiently long enough to trigger a response polarity, it should ultimately lead to organ curvature. This clinorotation effect should be observed with other clinorotation configurations as well, for similar reasons.

In this manuscript, we use *Arabidopsis thaliana* and *Brachypodium distachyon* seedling roots to test this hypothesis. We show that low-speed clinorotation under a variety of configurations triggers root tip curvature in which direction and degree are dictated by the polarity and period of rotation, respectively. Under vertical clinorotation, seedling roots responded to clockwise or counterclockwise clinorotation at speeds below 0.5 RPM by curving to the right and left, respectively, as long as the seedlings started clinorotation at a vertical orientation (Figure 1).

In the two-chamber vertical clinostat system described in Appendix A, the seedlings are initially positioned upright in one of the chambers, and upside down in the second chamber. When clinorotated clockwise, the seedling roots in the first chamber curve to the right, whereas those in the second chamber curve to the left (Appendix A). Interestingly, during the first half-turn of clockwise clinorotation, the right flank of the seedlings is directed toward gravity in chamber 1 and away from it in chamber 2, again potentially explaining the opposite polarities of root tip curvature displayed by these seedlings at the end of the experiment. This conclusion seems reinforced by our observation of opposite responses when counterclockwise clinorotation is applied (Appendix A).

It should be noted here that the angles of curvature reached by the roots of chamber 2 seedlings exposed to 0.2 RPM clinorotation are smaller than those displayed by chamber 1 seedlings (Appendix A). The reason for this difference is not clear, although it is possible that the mechanostimulus associated with quickly turning the plates 90 degrees to position them within chamber 2 before clinorotation might somewhat alter their ability to respond to gravistimulation [43].

When plates are positioned parallel to the clinostat axis, with *Brachypodium* seedlings oriented upright at the beginning of clinorotation (parallel clinorotation, Appendix A), forward clinorotation at 0.2 RPM results in directing the air-facing side of the seedlings to the gravity vector during the first half-turn, ultimately leading to the roots lifting off the agar surface. When clinorotated in the opposite direction, the roots tend to either penetrate the agar surface, or meander on it if they cannot penetrate (Appendix A). Similar results are obtained when *Brachypodium* seedlings are positioned horizontally on the clinostat (Appendix A).

In all these low-speed clinorotation experiments, the roots curve toward the side that faces the gravity vector during the first half-turn of clinorotation, as long as they start in a vertical orientation and the speed of rotation is sufficiently low. Indeed, 3 RPM clinorotations do not trigger root tip curvature under these experimental conditions (Figure 4). Therefore, our results are compatible with a role for gravitropism in modulating the low-speed clinostat effect on root growth direction.

If gravitropism is an integral component of the clinostat effect on root growth direction, mutations that affect gravitropism should alter the clinostat effect on root growth direction. Furthermore, the clinostat effect should be associated with the formation of a lateral auxin gradient across the root tip, a process accompanied by relocalization of the PIN3 and PIN7 auxin efflux facilitators to the lateral membrane of columella statocytes early during clinorotation (reviewed in [1]). Our results are consistent with both assumptions. Indeed, mutations that affect early phases of gravity sensing and signal transduction in *Arabidopsis thaliana* strongly affect root tip responses to low-speed clinorotation (Figure 2, Appendix A). This includes a mutation in the *PGM* gene, which encodes phosphoglucomutase, an enzyme that contributes to starch biosynthesis. Mutations in this gene have been reported to affect starch content in the amyloplasts of plant statocytes, thereby affecting their ability to sediment in response to gravistimulation [13]. The defect in mutant amyloplast sedimentation correlated with altered gravitropic response in roots, hypocotyls, and stems of *Arabidopsis*, in agreement with a role for amyloplast sedimentation in gravity sensing [44]. Importantly, previous studies of amyloplast sedimentation in the root cap statocytes of wild type *Arabidopsis thaliana* seedlings under the same growth conditions as our study have demonstrated significant sedimentation after only 1 or 2 min of reorientation [8], which is compatible with the time needed for the clinostat to complete its first half-turn under the inductive conditions used in this project. Taken together, these results are compatible with a role for amyloplast sedimentation, during the first half-turn of clinorotation (due to plant reorientation within the gravity field), in clinostat-induced root tip curvature.

Mutations in *ARG1* are also shown to alter root tip responses to low-speed clinorotation in this project, a phenotype that is suppressed by expressing a wild type transgene in mutant plants (Figure 2). *ARG1* and its paralog *ARL2* encode J-domain membrane-associated proteins that are essential for PIN relocalization to the bottom membrane of root cap statocytes and their cytoplasmic alkalinization in response to gravistimulation [7], thereby modulating very early steps of gravity signal transduction in the statocytes. The impact of *arg1* mutations on root tip curvature responses to low-speed clinorotation is again consistent with a role for gravitropism in this process.

Mutations in the *TOC132*/*MAR2* gene were previously shown to strongly affect root gravitropism in an *arg1-2* mutant background while having little to no effect in a wild type *ARG1* background. This strong genetic interaction between *TOC132*/*MAR2* and *ARG1* indicates a role in gravity signal transduction within the statocytes that does not impact starch accumulation or amyloplast sedimentation [8]. *TOC132*/*MAR2* encodes a component of the translocon of the outer membrane of chloroplasts complex located on the surface of plastids, which contributes to protein import from the cytoplasm into the plastids and impacts gravity signal transduction [8,45]. In the experiments described in this manuscript, mutations in this gene strongly affected root tip curvature responses to low-speed clinorotation despite being present in a wild type *ARG1* background. This result suggests that the root curvature response to clinorotation may constitute a more sensitive assay for gravitropism than a simple reorientation assay.

Considering that root gravitropism typically involves the creation of a lateral auxin gradient across the root cap, followed by its propagation toward the elongation zone where it promotes differential cellular elongation and tip curvature, we used a transgenic DII-VENUS auxin reporter to document the formation of a lateral auxin gradient across clinorotated seedlings, showing signal disappearance on the flank of the root that was directed toward gravity during the first half-turn of rotation. This clinostat-induced signal gradient occurred within 1 h of clinorotation and decreased after 2 and 6 h. It occurred under low-speed clinorotation (0.2 RPM), but was not observed under non-inductive conditions at 3 RPM or in seedlings that were not clinorotated. Its intensity was lower than the sharp gradient observed in response to direct gravistimulation (Figure 5; Appendix A). Interestingly, the PIN3 protein was also shown to relocalize to the side of the columella cells after 1 h of 0.2 RPM clinorotation. This relocalization resulted in PIN3 accumulation on the side membrane of the statocytes that was directed toward gravity during the first half-turn of clinorotation. It was less pronounced than observed in response to direct gravistimulation, and it was not observed under 3 RPM rotation (Figure 5; Appendix A).

Taken together, these results are compatible with the first half-turn of clinorotation at 0.2 RPM providing sufficient directional information to polarize the root cap statocytes, leading to PIN protein relocalization in the statocytes, auxin gradient formation along the root tip, and curvature response. The levels of PIN3 relocalization and steepness of the lateral auxin gradient triggered by 0.2 RPM clinorotation were not as strong as those established upon direct gravistimulation because the latter provides a continuous stimulus over time, with progressively decreasing inclination, whereas the stimulus provided during clinorotation occurs only for a limited time (half the period of clinorotation) before being interrupted by opposite inclination. The observation of increased curvature angles generated by clinorotation of increased periodicity is compatible with this explanation (Figure 3).

We have been able to also show a logarithmic relationship between angle of curvature and period of clinorotation. When extended to the time axis, this function allows the estimation of a parameter we called realization time (T_R_), which we defined as the minimal period of half-turn clinorotation needed to set up a polarity of tip curvature response to clinorotation. While this approach was inspired from the presentation time analysis carried out to evaluate gravisensitivity of plant organs, it is important to recognize that the realization time parameter differs from the presentation time because in T_R_ analysis, the same stimulus is provided at each following inductive cycle during clinorotation, whereas a single limited gravistimulus is provided prior to horizontal clinorotation in presentation time analysis. In this sense, the realization time is more reminiscent of the perception time parameter, defining sensitivities based on repeated short periods of gravistimulation. This conclusion is supported by the observation of a need for a minimum of four cycles of inductive clinorotation to allow the development of a significant curvature response at the end of the experiment (Figure 4). Future experiments will be aimed at elucidating the relationship existing between curvature response and total stimulation to evaluate possible reciprocity between number of inductive cycles and periodicity of each cycle as components of stimulation. Experiments have also been initiated to elucidate the potential impact of different speeds of clinorotation during the response (non-inductive) phase on the final curvature.

Finally, it should be noted that our experiments were carried out with rather long periods of clinorotation, allowing for better reproducibility of the results (12 to 18 h). However, this implies that significant auto-straightening is likely to have occurred during such long periods of clinorotation, considering that the first signs of curvature were visible after 5 h (Appendix A). Current research is aimed at documenting the impact of proprioception on the curvature responses to clinorotation using biometric data calculated from time-lapse images recorded using an infrared-sensitive camera mounted on the clinostat (Su et al., unpublished work).

Previous reports documenting plant responses to clinorotation at different speeds demonstrated effects on organ curvatures only at higher speeds, where the organ tip appeared to reorient following the force vector resulting from both gravity and centrifugal acceleration. Clinorotation at the low speed of 1 RPM (or even 0.3 RPM in one case), did not trigger organ curvature responses [24,34,46,47,48,49]. It should be pointed out that the experiments we describe here also did not trigger any responses at 1 RPM. However, clear curvature responses were observed to clinorotations at 0.5 and 0.2 RPM. We also note that different accessions of *Brachypodium distachyon* displayed dramatically distinct responses to 0.2 and 0.5 RPM clinorotations, with Bd21-3 showing strong responses at 0.2 RPM and a weaker response at 0.5 RPM, whereas Ron-2 displayed no responses at all (Figure 6). Similarly, mutations affecting early steps of gravity sensing and signal transduction in *Arabidopsis thaliana* affected root curvature responses to inductive clinorotation (Figure 2). On the other hand, wild type *Arabidopsis* seedlings of the Col and Ws ecotypes were previously shown to display significant curvature responses to clinorotation at 1 RPM under distinct experimental conditions [37], whereas they showed no curvatures at 1 RPM in the experiments described here. We conclude that the genetic background of the plants being analyzed and the environmental and/or experimental conditions used to carry out these experiments will affect root tip curvature responses to clinorotation, emphasizing the need to carefully report both.

In conclusion, our observations suggest a new method to evaluate root gravitropism, and possibly gravisensitivity, using clinorotation. We are currently taking advantage of the large variability existing between *Brachypodium distachyon* accessions to identify molecular mechanisms that contribute to this variability using genome-wide association studies, hoping to uncover novel mechanisms that contribute to gravity sensing and signal transduction in roots.

## 4. Materials and Methods

### 4.1. Plant Material and Growth Conditions

Seeds of *Arabidopsis thaliana* accessions Col and Ws (originally received from ABRC, Columbus, OH, USA) and of *Brachypodium distachyon* Bd21-3 and Ron-2 accessions (received from the Amasino laboratory at the University of Wisconsin–Madison) were used as experimental materials in this study. *Arabidopsis* mutants with defects in gravity signal transduction were also used in this project, including *arg1-2* (Ws background) and *arg1-3* (SALK_024542, Col background) [6,7,8], *pgm-1* (Col background) [8], *toc132-3* (GABI_394E01, Col background) [50], *toc 132-4* (SALKseq061933), and *mar2-1* (Ws background) [6,8]. A transgenic *Arabidopsis thaliana arg1-2* mutant expressing the *35Sp:ARG1* transgene (*arg1-2[35S:ARG1]*) was also used as a control [7]. Finally, transgenic lines expressing the *DII::VENUS* and *pPIN3::PIN3-GFP* reporters in the Col background were used in confocal microscopy [5,22]. Seedlings from the Col accession were chosen for most experiments because their roots display a minimal amount of root skewing on hard agar surfaces [37], a behavior that could potentially complicate interpretation of the results.

*Arabidopsis thaliana* seeds were surface-sterilized, germinated, and grown on the surface of a half-strength LS medium with 0.8% agar under 16/8 h day/night cycles at 20–22 °C (cool-white LED light at ~70 µE·m^−2^·s^−1^), as described in [8].

*Brachypodium distachyon* seeds were dehusked and surface-sterilized with five successive 1 min washes in 95% (*w*/*v*) ethanol, and then air dried in a sterile hood for at least 20 min. Sterilized seeds were soaked with autoclaved Milli-Q water in darkness at 4 °C for 24 h. Seeds were then plated on the surface on 1% agar plates with 0.5× Murashige and Skoog medium (Sigma) and grown under red LED light at ~15 µE·m^−2^·s^−1^ (Tangkula Inc., Ontario, CA, USA) for 24 h to promote germination. Germinated seeds were transferred to new agar plates and grown under white LED light at ~75 µE·m^−2^·s^−1^ (Tangela Inc., Ontario, CA, USA) for 5 h to stabilize the transplanted seedlings. Then, the plates were set into the clinostat chambers in the dark for an additional 4 h before clinorotation assay.

### 4.2. Clinorotation Assays

After germination using the protocols described under the subsection *Plant Materials and Growth Conditions*, germinated seedlings were transferred to fresh 0.5× LS with 0.8% agar-based medium, aligned at the middle of the plate with the root tips directed downward. The new plates containing these transplanted seedlings were then mounted in the clinostat chamber with root tips oriented downward, and were maintained in this position for 4 h to stabilize the transplanted seedlings before clinorotation (unless stated otherwise). After stabilization, the seedlings were subjected to either vertical or parallel clinorotation.

In all but one of the clinorotation systems described below (the two-chamber vertical clinorotation system being the exception), the clinostat used an ATmega328 microcomputer to control a 12 V 350 mA stepper motor connected to a belt and gears. These were connected to a shaft with centrally mounted lightproof acrylic boxes to hold square Petri plates.

For **vertical clinorotation**, the clinostat chamber was positioned such that the plates it contained were perpendicular to the rotation axis (Appendix A). Half of the plates were oriented such that their lids were directed to the right of the clinostat, the other half with the lids directed to the left. This allowed us to test the effects of both clockwise and counterclockwise clinorotations on seedling root growth in the same experiment. In all experiments described in this manuscript, the direction of clinorotation (clockwise vs. counterclockwise) is defined based on observations from the back side (through the agar-based medium). In some vertical clinorotation experiments, half of the plates were transferred into the clinostat chamber with seedlings oriented vertically downward (roots directed upward) so that their roots would be exposed to gravistimulation from the other side during the first half-turn of clinorotation relative to those starting the experiment in the normal upright position. In this set of experiments, clinorotation was initiated immediately after transferring the plates into the clinostat chamber.

A **two-chamber-based vertical clinorotation** system was also designed to evaluate the clinostat effect on root tip curvature. The two-chamber-based clinostat was built by modifying a Thomas Scientific HYBAID hybridization oven. In this system, two chambers were located on opposite sides of the horizontal axis of the clinostat. The first chamber was loaded with plates carrying seedlings in an upright position, whereas chamber 2 was loaded with plates in the opposite orientation (upside down seedlings at the start of clinorotation; see Appendix A). In both chambers, half of the plates were inserted with their lids directed toward the left of the clinostat (clockwise rotation), the other half with the lids directed toward the right (counterclockwise rotation).

In **parallel clinorotation**, the plates were positioned parallel to the axis of rotation within the clinostat chamber, with their seedlings oriented **parallel to the gravity vector** before clinorotation (Appendix A). The plates were positioned back-to-back in the chamber, with half of them having their lids directed toward the front of the chamber, and the other half having lids directed toward the back of the chamber. Therefore, during the first half-turn of clinorotation, the side of the roots facing the front of the plate was directed toward gravity in the first group of plates, whereas the side of the roots facing the medium was directed toward gravity in the second group of plates (Appendix A).

**Horizontal clinorotation** was carried out by positioning the clinostat chamber in such a way that both the plates and the seedlings within them were parallel to the axis of rotation (Appendix A). Here again, the plates were positioned back-to-back in the chamber, exposing seedling roots to gravity on the side exposed to air in the first group of plates, and on the side facing the medium in the second group, during the first half-turn of clinorotation.

After clinostat loading and a resting period (when applicable), the seedlings were clinorotated in darkness for 12 or 18 h (for *Brachypodium* and *Arabidopsis* seedlings, respectively) at the speeds defined in the text. At the end of clinorotation, the plates were scanned with an Epson Perfection V33 scanner, and root tip angles were measured using ImageJ. Rose plots were generated by Microsoft Excel. To generate publication-quality pictures, we used Photoshop to increase the brightness and contrast of representative images, avoiding any shape distortion.

### 4.3. Microscopy

Transgenic Col seedlings expressing the *DII::VENUS* and *PIN3p::PIN3-GFP* reporter constructs were grown for 5 days on the surface of 0.5 × LS, 0.8% agar (type E, Sigma) medium with a coverslip embedded in the center of the plates. Plates were then mounted into a clinostat and stabilized for more than 4 h in the dark to avoid mechanical stimuli. After different periods of clinorotation (defined in the text), a coverslip was placed on top of the seedlings, and the mount was transferred to the confocal microscope stage for imaging. This sandwiching approach allowed us to unambiguously relate reporter signal asymmetry to clinorotation polarity for each sample. Images were taken with a Zeiss LSM 780 confocal microscope using excitation and emission wavelengths of 514 and 570 nm to detect VENUS, or 488 and 546 nm to detect PIN3-GFP. The detector gain range was set up for 730 to 850 nm and the scan mode was “by frame”. Seedlings were rotated clockwise but imaged from either side of the coverslip depending on sample setup. Signal intensity was measured using ImageJ (Plot Profile). A heatmap was generated by inputting the grey image of the signal channel and using the Look-Up Tables (LUTs) plug-in from ImageJ to reflect the difference in intensity by color [51].

## Figures and Tables

**Figure 1 ijms-24-01540-f001:**
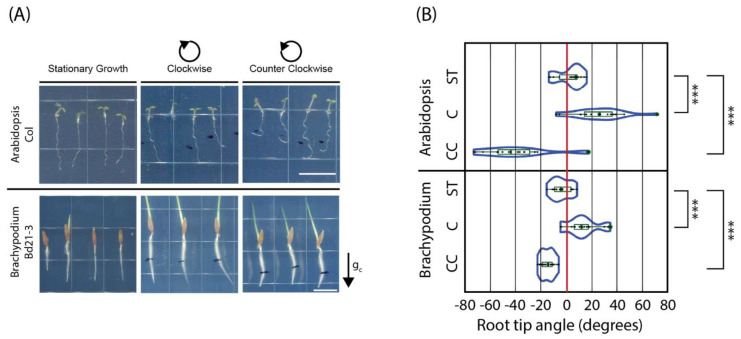
Representative root tips of *Arabidopsis thaliana* and *Brachypodium distachyon* seedlings curve in response to low-speed vertical clinorotation: (**A**) 5-day-old *Arabidopsis* (Col, upper panel) and 3-day-old *Brachypodium* (Bd21-3, lower panel) seedlings were clinorotated clockwise (C) or counterclockwise (CC) at 0.2 RPM or stationary growth (ST) in darkness for 24 h. Rotation direction is defined based on observations from the back of the plates, through the agar-based medium; g_c_ refers to the gravity vector at the onset of clinorotation; white bars refer to 1 cm. (**B**) Quantification of root tip angles at the end of clinorotation (*n* = 12 to 17). The distributions of root tip angles for each experiment are represented as violin plots. The box plots within each violin indicate median (horizontal bar) and mean values (X), along with the quartiles (rectangular boxes) and standard deviations (whiskers) of the distributions. *** Refers to the *t*-test *p*-value of < 0.001. These experiments were repeated twice, with similar results.

**Figure 2 ijms-24-01540-f002:**
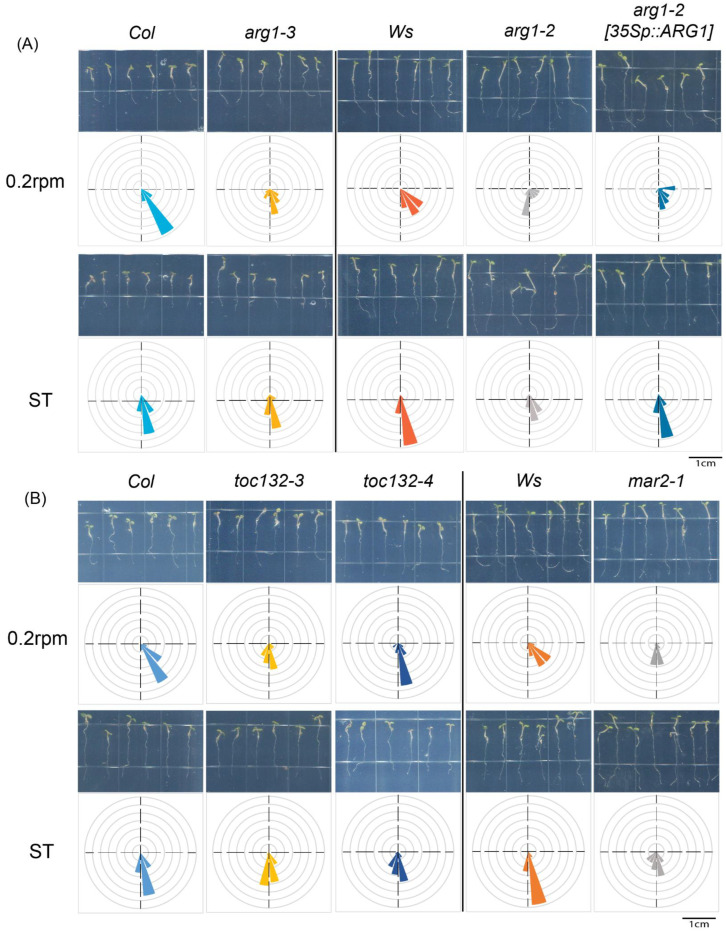
Gravitropism-deficient *Arabidopsis* mutants do not show root tip curvature responses to low-speed clinorotation: 5-day-old wild type, mutant, and transgenically rescued *Arabidopsis* seedlings were subjected to either 0.2 RPM vertical clinorotation or stationary growth (ST) in darkness for 18 h. Rose plots of root tip angles measured at the end of each experiment are represented for each genotype (represented at the top) and each experimental condition (represented on the left). These rose plots are split into 18 bins of 20 degrees each. The proportion of the measured population of root tip angles falling into a specific bin is represented by a wedge within that bin. The bin representing seedlings with root tips directed toward the bottom within 0 to 20 degrees of the vertical is directed to the bottom, the bin representing 80–100 degrees is directed to the right, the one representing 160–180 degrees to the top, and the one representing 260–280 degrees to the left. Photographs of representative seedlings taken at the end of each experiment are shown above each rose plot. Several alleles of (**A**) *arg1* and (**B**) *toc132* were tested in this experiment, along with their respective wild types, and an *arg1-2* mutant line transformed with a *35Sp::ARG1* transgene (*arg1-2[35S:ARG1]*). *n*N = 12–25.

**Figure 3 ijms-24-01540-f003:**
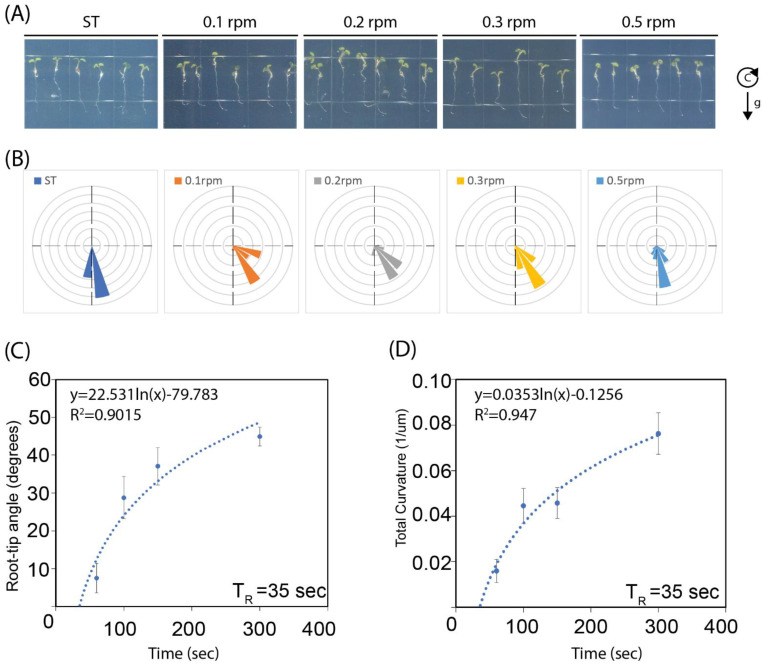
The speed of clinorotation affects the level of root tip curvature: (**A**) Representative root tip images of *Arabidopsis* seedlings clinorotated clockwise at the indicated speeds (0.1, 0.2, 0.3, and 0.5 RPM) and stationary growth (ST) in darkness for 18 h. g_c_ represents the gravity vector at the onset of clinorotation. (**B**) Rose plot of the root tip angles (in degrees) at the end of clinorotation at the indicated speeds. (**C**,**D**) Root tip angles in degrees (**C**) and total root tip curvatures (**D**) reached at the end of 18 h clinorotation as functions of the duration of the first half-turn of clinorotation (half periodicity, in s). Best-fit logarithmic models (dotted lines) are represented on these graphs, whose functions are indicated at the top along with the corresponding correlation values (R^2^). The realization time (T_R_) deduced from these functions is indicated at the bottom right of each graph. *n*= 20. This experiment was repeated twice, with similar results.

**Figure 4 ijms-24-01540-f004:**
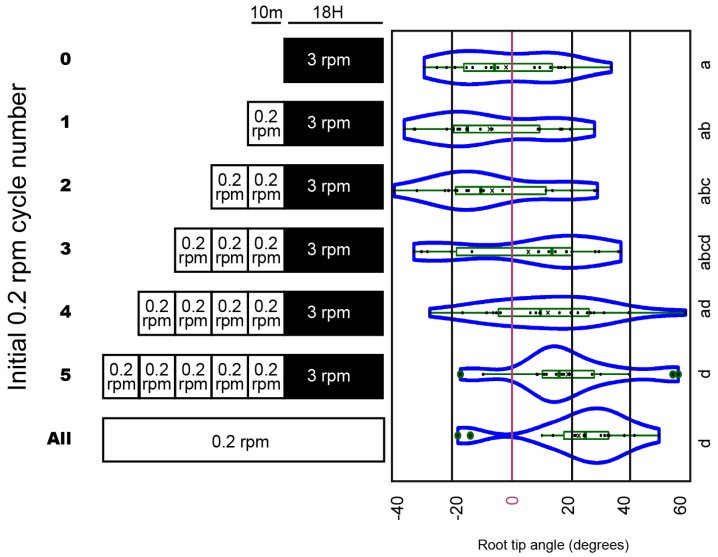
Several rounds of inductive clinorotation are needed for maximal root tip curvature response: 5-day-old wild type *Arabidopsis thaliana* seedlings (Col accession) were subjected to 0, 1, 2, 3, 4, or 5 rounds of inductive 0.2 RPM clinorotation in darkness before being exposed to 18 h non-inductive clinorotation at 3 RPM. Root tip angles were measured at the end of each experiment and plotted over the number of inductive clinorotation rounds. The results are represented in the form of a violin chart. The box plots within each violin indicate median (horizontal bar) and mean values (X), along with the quartiles (rectangular boxes) and standard deviations (whiskers) of the distributions. One-way ANOVA analysis was performed. Different letters identify significantly different responses (Tukey’s HSD test, *p* < 0.05). This experiment was repeated twice, with similar results.

**Figure 5 ijms-24-01540-f005:**
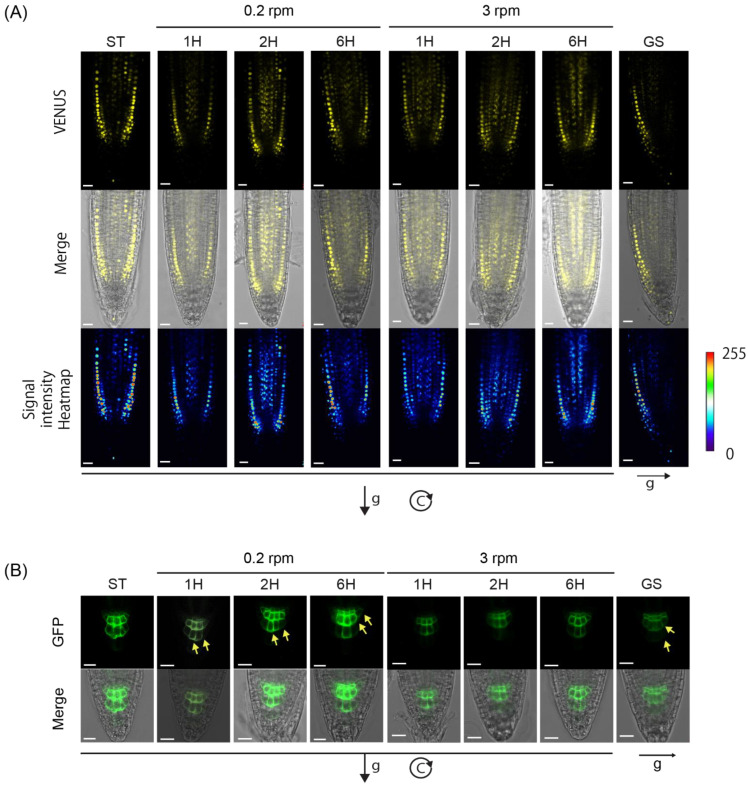
Low-speed vertical clinorotation leads to asymmetric auxin distribution across the root tip: 5-day-old transgenic *Arabidopsis thaliana* seedlings (Col) expressing the *DII::VENUS* auxin reporter were subjected to 0.2 RPM (**left**) or 3 RPM (**right**) vertical clinorotation in darkness for the time periods indicated above each picture. Non-clinorotated (ST) and constantly gravistimulated (GS) controls were included in this experiment. Panel (**A**) shows representative confocal images of root tip DII::VENUS fluorescence signals under the rotation conditions and times defined at the top; the middle images show the confocal DII-VENUS fluorescence signals merged with bright field views of the same roots; the bottom images show the results of signal quantification in the form of signal intensity heatmaps generated using the LUT plug-in from the ImageJ. Scale bars represent 50 µm. Panel (**B**) demonstrates how low-speed clinorotation leads to mild PIN3 relocalization in the columella cells of the root cap. 5-day-old transgenic *Arabidopsis thaliana* seedlings (Col accession) expressing a PIN3-GFP construct were either clinorotated at 0.2 and 3 RPM in darkness, gravistimulated, or grown straight downward without clinorotation or gravistimulation (ST) for 0, 1, 2, or 6 h. The PIN3-GFP fluorescent signal was then analyzed by confocal microscopy. A mild polarization of the signal appeared in the columella cells of the root cap (yellow arrows) within 1 h of 0.2 RPM clinorotation and continued over a period of 6 h. Similar polarization was not seen upon 3 RPM clinorotation or when the seedlings were neither gravistimulated nor clinorotated. Quantification of VENUS signals from panel (**A**) were presented in Appendix A. In panels (**A**) and (**B**), the direction of gravity (g) at the onset of clinorotation (or gravistimulation, GS) is represented by a black arrow.

**Figure 6 ijms-24-01540-f006:**
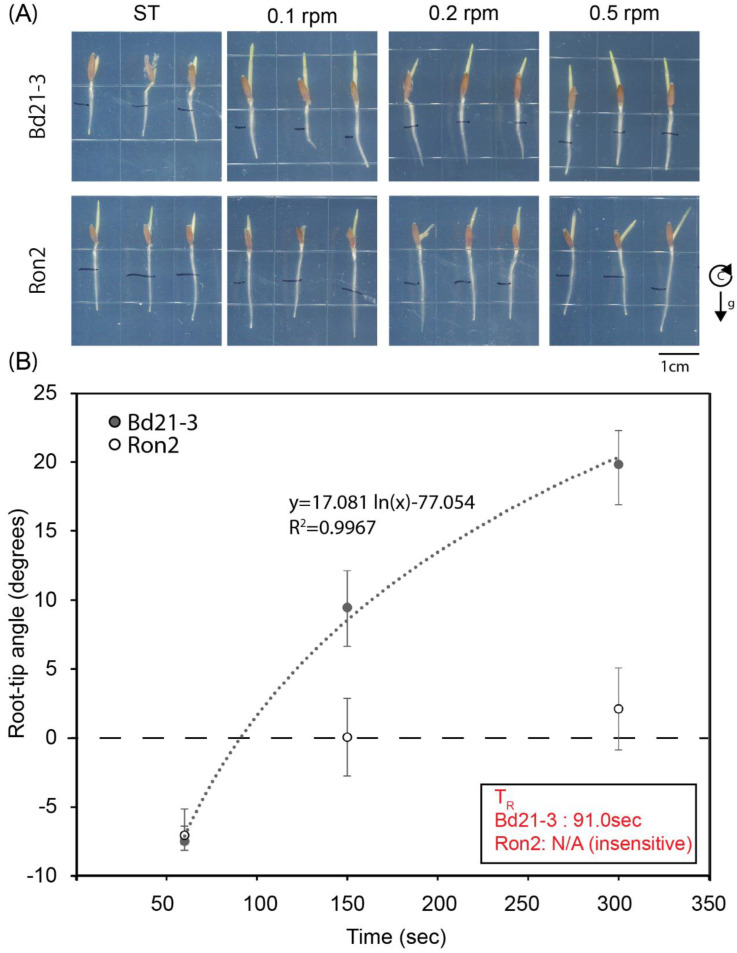
*Brachypodium distachyon* accessions display distinct root tip curvature responses to low-speed clinorotation: 3-day-old *Brachypodium distachyon* seedlings of the Bd21-3 and Ron-2 accessions were subjected to clinorotation at speeds of 0 (ST control), 0.1, 0.2, and 0.5 RPM for 14 h in darkness. The seedlings were photographed at the end of each experiment. Panel (**A**) shows representative root tip images of Bd21-3 and Ron-2 seedlings exposed to the clinorotation conditions summarized at the top. (**B**) Averaged root tip angles at the end of clinorotation along with their standard deviations are plotted over the time needed to complete half a turn of clinorotation in the corresponding experiment. A logarithmic model was fit to the data (dotted line) and used to calculate the realization time (T_R_) for responding seedlings. The function associated with the best-fit model is represented at the top of the graph along with the corresponding R^2^ value. Ron-2 seedlings did not respond to clinorotation, thus it was not possible to assign a specific T_R_ value to them. *n* = 6–11. This experiment was repeated twice, with similar results.

**Table 1 ijms-24-01540-t001:** Quantification of root-tip curvature responses to stationary growth (ST) or 0.2 rpm clinorotation for wild type (Col and Ws) and gravitropism-deficient Arabidopsis mutants growing on agar surfaces. Mean root-tip angles and standard deviations are shown along with the p-values associated with statistical analyses of the significance of difference between samples. Panels (A) and (B) summarize the results of two independent experiments.

				*p*-Value		
Genotype	Clinorotation	Mean of Root Tip Angle	Stdev	*t*-Test ST vs. 0.2 rpm ^a^	*t*-Test Mutant vs. WT ^b^	F-Test Mutant vs. WT ^c^
** *Col (WT)* **	0.2 rpm	33.35	13.74	3.30 × 10^−9^		
	ST	10.1	14.36			
** *arg1-3* **	0.2 rpm	10	28.89	0.47	5.30 × 10^−4^	
	ST	2.96	21.93		0.18	0.048
** *toc132-3* **	0.2 rpm	2.41	34.65	0.79	1.40 × 10^−4^	
	ST	0.34	15.42		0.01	0.659
** *toc132-4* **	0.2 rpm	7.75	24.56	0.06	4.10 × 10^−5^	
	ST	−4.32	19.17		5.16 × 10^−4^	0.089
** *Ws (WT)* **	0.2 rpm	43.37	28.5	2.70 × 10^−4^		
	ST	7.92	12.84			
** *arg1-2* **	0.2 rpm	14.6	33.46	0.84	3.30 × 10^−3^	4.20 × 10^−2^
	ST	2.96	24.74		0.59	
** *arg1-2[35S:ARG1]* **	0.2 rpm	40.95	34.99	0.01	0.81	
	ST	8.78	14.41		0.88	0.709
** *mar2-1* **	0.2 rpm	−5.52	27.15	0.99	2.10 × 10^−7^	
	ST	−5.62	23.3		0.09	0.058
** *Col (WT)* **	0.2 rpm	15.34	11.51	3.88 × 10^−3^		
	ST	4.12	11.24			
** *pgm1* **	0.2 rpm	−0.15	30.13	0.99	4.26 × 10^−2^	
	ST	−0.02	18.94		0.41	0.028

^a^*t*-Test between stationary growth (ST) and clinorotation (0.2 rpm) within the same genotype. ^b^*t*-Test between each mutant and corresponding wild-type within the same treatment. ^c^ F-Test between each mutant and corresponding wild-type in stationary growth (ST) condition.

## Data Availability

Not applicable.

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
