# Peer review of "Low-Speed Clinorotation of Brachypodium distachyon and Arabidopsis thaliana Seedlings Triggers Root Tip Curvatures That Are Reminiscent of Gravitropism"

_ijms, 2023, doi:10.3390/ijms24021540_

Round 1

Reviewer 1 Report

The manuscript describes a set of experiments aimed to understand the effect of rotation to the development of the root tip.  

A couple of concerns:

Figure 3 C and D provide models quantitatively describing the relationship between the speed of rotation and thecurvature of the tip.  The models use only three different rotation speeds - so the resulting high fit of the models is likely explained by a very small number of points used in this analysis.  the same critic is applicable to the experiment describe by Figure 6.

Figure 4 is likely not necessary as it contains a large number of p-values - information that could be included in the text (the same goes with Figure 5 C).

The expression patterns ofFigure 5 are remarkable and the figure ahowsnicely expression patterns of the correcposning proteins.

Author Response

Dear Editor and Reviewers,

Thank you for reviewing our manuscript and offering feedback and suggestions. Below, we address the various points raised by each reviewer on the first draft of our manuscript, and specify the modifications we made in the manuscript to address these points. The reviewers comments are provided in bold characters at the top of each section, and our replies are in regular font, red characters. We believe that the changes made in response to the reviewers’ feedback have substantially improved the quality of our manuscript, and we want to thank them for their excellent comments and suggestions.

Reviewer #1

The manuscript describes a set of experiments aimed to understand the effect of rotation to the development of the root tip.  

A couple of concerns:

Figure 3 C and D provide models quantitatively describing the relationship between the speed of rotation and the curvature of the tip.  The models use only three different rotation speeds - so the resulting high fit of the models is likely explained by a very small number of points used in this analysis.  The same critic is applicable to the experiment describe by Figure 6.

This is a very good point. However, equipment limitations prevent us from testing more than 4 different rpm conditions in the same experimental set up. We want and need to run these experiments in parallel (at the same time) to avoid complications from experimental variability. Therefore, to address this reviewer’s comment, we repeated the experiment described in Figure 3 with one additional data point (0.3rpm). The results from this new experiment involving four data points show a logarithmic model still fits nicely with the observed data (R2 values above 0.9 for both root-tip angle and total root-tip curvature parameters).  The novel Tr value calculated from this new dataset is 35 sec, very similar to the value we previously obtained in experiments involving only 3 data points.  We now show the results of this new experiment involving 4 data points in Figure 3. We also modified Figure 3 to include the results of a stationary growth control (which we always include in our experiments; see below).

Regarding Figure 6, we unfortunately didn’t have enough Ron2 seeds to repeat this experiment with 4 data points.  However, in view of the results discussed above for Figure 3, we believe the results with 3 data points are sufficient to reach the rather simple conclusion we draw from this analysis: Bd21-3 responds to clinorotation by root tip curvature; Ron2 does not under these conditions.

Figure 4 is likely not necessary as it contains a large number of p-values - information that could be included in the text (the same goes with Figure 5 C).

Both reviewers made similar suggestions regarding these figures, and we agree. Therefore, we moved panels B and C from Figure 5 into Supplemental Figure S7 (becoming Supplemental Figure S7, panels A and B).  We also moved the PIN3 localization images from Supplemental Figure S7A into Figure 5 (becoming Figure 5, panel B) to address one of  reviewer #2’s suggestions.

For figure 4B, we removed the table with statistics from this Figure, and replaced it with letter symbols within the graph to indicate statistically significant groups based on an ANOVA analysis.

The expression patterns of Figure 5 are remarkable and the figure shows nicely expression patterns of the corresponding proteins.

Thank you for the kind note.

We would like to once again thank the reviewers for their excellent suggestions. We believe the changes we made to this manuscript in response to their feedback improved the quality of this document, and hope it is now ready for publication.

Reviewer 2 Report

In this manuscript, Su et al. report the effects of slow-speed clinostat setups on root gravitropic response of Arabidopsis thaliana and Brachypodium distachyon. They showed that gravitropism contributes to the clinostat effect on the direction of root growth curvature under low speed clinorotation. They also studied auxin distribution as well as the localization of PIN3 protein, an indispensable auxin efflux carrier for gravitropism.  They found that low-speed clinorotation causes an asymmetric auxin gradient and results in a relocalization to the lower side of the columella cells of PIN3. Gravity is a critical environmental factor affecting the morphology and function of plants on Earth. As the clinorotation is the most feasible way to mimic microgravity on Earth, this report is fundamental but yet important.

However, there are some issues that the authors should address. These issues are listed below:

Figure 1.

1)Better quality images of the Arabidopsis and Brachypodium seedlings should be provided. How many root tips did you measure? Please increase the number of seedlings for each sample analyzed.  How many technical repeats did the authors perform?

2)The corresponding Arabidopsis and Bd21-3 seedlings grown under straight growth conditions should have been included and presented.

Figure 2

1)Better quality images of the Arabidopsis roots should be provided.

2) the Authors state(ln252) that each plot is divided into 20 degree sections. What do you mean by that? Did you mean 90-degree? Please explain.

3) In my experience, the root length of the 5-day-old Arabidopsis seedlings grown under the 16h/8h light/dark cycle should be approximately 2-2.5cm. Which are the root lengths of the seedlings of each sample? Does clinorotation affect root growth?

Table 1

In my opinion the statistical analysis belongs to the Supplementary material. The quantification of root-tip   curvature responses could be presented as a chart.

Figure 3

Better quality images of the Arabidopsis seedlings should be provided.

Figure 4

In my opinion the first panel of Figure 4A belongs to Supplementary material. The violin plot should remain. Please add the corresponding p-values to violin plot.

Figure 5

5A. What does the arrow of g indicate in GS treatment?

5C. The table belongs to Supplementary material.

I strongly suggest to move Sup6A to main Figure 5.

Lns 392-395. Please discuss the biological significance of your finding

Author Response

Dear Editor and Reviewers,

Thank you for reviewing our manuscript and offering feedback and suggestions. Below, we address the various points raised by each reviewer on the first draft of our manuscript, and specify the modifications we made in the manuscript to address these points. The reviewers comments are provided in bold characters at the top of each section, and our replies are in regular font, red characters. We believe that the changes made in response to the reviewers’ feedback have substantially improved the quality of our manuscript, and we want to thank them for their excellent comments and suggestions.

Reviewer #2

In this manuscript, Su et al. report the effects of slow-speed clinostat setups on root gravitropic response of Arabidopsis thaliana and Brachypodium distachyon. They showed that gravitropism contributes to the clinostat effect on the direction of root growth curvature under low speed clinorotation. They also studied auxin distribution as well as the localization of PIN3 protein, an indispensable auxin efflux carrier for gravitropism.  They found that low-speed clinorotation causes an asymmetric auxin gradient and results in a relocalization to the lower side of the columella cells of PIN3. Gravity is a critical environmental factor affecting the morphology and function of plants on Earth. As the clinorotation is the most feasible way to mimic microgravity on Earth, this report is fundamental but yet important.

However, there are some issues that the authors should address. These issues are listed below:

Figure 1.

  • Better quality images of the Arabidopsis and Brachypodium seedlings should be provided. How many root tips did you measure? Please increase the number of seedlings for each sample analyzed.  How many technical repeats did the authors perform?
  • The corresponding Arabidopsis and Bd21-3 seedlings grown under straight growth conditions should have been included and presented.

We increased the sample size from a minimum of 8 seedlings to a minimum of 15 seedlings per treatment in this experiment to address this request, and also added symbols indicating statistical significance to the graph shown in panel B. Please note that the differences in responses between treatments are statistically highly significant, indicating the sample size is adequate for this analysis.

All the experiments discussed in this paper have been repeated at least twice to make sure they are reproducible, and the data shown in the figures are representative of all observations. Unfortunately, we had omitted mentioning this point in the figure legends in the original draft. This has been corrected in the current draft.

The original images for the root photos had to be scanned at 300-dpi resolution, directly on the agar plates, to allow subsequent analysis of tip angles using ImageJ. Therefore, these images are the best we can provide. While the Arabidopsis roots are less visible that Brachypodium’s, we believe they remain visible on our image files. If the reviewer still believes this is not sufficient, we can modify the background settings using Photoshop and specify the setting changes in the legend. However, we feel uncomfortable doing so.

Figure 2

  • Better quality images of the Arabidopsis roots should be provided.

Please see our reply above to a similar suggestion for figure 1.

  • the Authors state(ln252) that each plot is divided into 20 degree sections. What do you mean by that? Did you mean 90-degree? Please explain.

As a histogram, the Rose plot provided in Figure 2 defines the number of seedlings with root tip angles falling within defined bins. The 360 degrees defining a full circle (from vertical down to horizontal to vertical up to horizontal to vertical down) are split into18 bins of 20 degrees each. Rather than representing the histogram in its usual linear format, the Rose plot illustrates it as a full circle (to better reflect the reality). The proportion of seedlings with root tip angles falling within a particular bin (say 0 to 20 degrees) is represented by a wedge. The larger the wedge, the higher the proportion of seedlings with root tip angles falling in that bin within the population analyzed. The grey circles surrounding the center of the plot represent defined fractions of the overall population (in 12.5% increments).

To clarify this, we added a sentence (represented in yellow below) in the legend to figure 2.

Lines 256 to Lines 273: “Figure 2. Gravitropism-deficient Arabidopsis mutants do not show root-tip curvature responses to low-speed clinorotation. 5-day-old wild-type, mutant, and transgenically rescued Arabidopsis seedlings were subjected to either 0.2 rpm vertical clinorotation or stationary growth (ST) in darkness for 18 hours. Rose plots of root-tip angles measured at the end of each experiment are represented for each genotype (represented at the top) and each experimental condition (represented at the left). These Rose plots are split into 18 bins of 20 degrees each. The proportion of the measured population of root tip angles falling into a specific bin is represented by a wedge within that bin. The bin representing seedlings with root tips directed toward the bottom within 0 to 20 degrees of the vertical is directed to the bottom. The bin representing 80-100 degrees is directed to the right, the one representing 160-180 degrees to the top, and the one representing 260-280 degrees to the left.  Photographs of representative seedlings taken at the end of each experiment are shown above each rose plot. Several alleles of (A) arg1 and (B) toc132 were tested in this experiment, along with their respective wild types, and an arg1-2 mutant line transformed with a 35Sp::ARG1 transgene (arg1-2 [35S:ARG1]).”

  • In my experience, the root length of the 5-day-old Arabidopsis seedlings grown under the 16h/8h light/dark cycle should be approximately 2-2.5cm. Which are the root lengths of the seedlings of each sample? Does clinorotation affect root growth?

Different growth environments will lead to differences in root growth rates.  In our own conditions, 5 day-old Arabidopsis seedlings are about 1 to 1.5 cm.  To test if clinorotation affects root growth, we measured root growth during the clinorotation period (18 hours) for each genotype, and the results are now shown in Supplemental figure S1. These results show no significant differences in root growth between clinorated (0.2rpm) and stationary seedlings (ST) for any of the genotypes tested in our work. To emphasize this observation, we also added the following sentences to the Results section:

Lines 186-187: “Clinorotation did not significantly affect the growth of wild type Col and Ws roots (Supplemental Figure S1).”

Lines 193-196: “The roots of stationary seedlings did not develop a tip curvature (Figure 1 A and B). Clinorotation did not significantly affect Brachypodium root growth relative to the stationary growth control (T-Test p value = 0.41 and 0.22).

Lines 294-296: “Clinorotation did not alter root growth in either wild type or mutant seedlings relative to stationary controls (Supplemental Figure S1, A-C).”

Table 1

In my opinion the statistical analysis belongs to the Supplementary material. The quantification of root-tip   curvature responses could be presented as a chart.

Table 1 is actually an additional quantification summary for figure 2.  We feel the complexity of the experiment presented in figure 2 justifies the use of Rose plots to display wild type and mutant root tip responses to clinorotation. Unfortunately, this reader-friendly visualization method does not allow an easy representation of statistical data, which are critical to the interpretation of these experiments. Therefore, we generated Table 1 to show the quantification results and a summary of their statistical analysis. We feel these results are critical to data interpretation in this project, and belong to the document itself (rather than being transferred to the Supplementary Materials). If the reviewer feels strongly about this suggestion, though, we’ll reluctantly move this table to the Supplementary Materials.

Figure 3

Better quality images of the Arabidopsis seedlings should be provided.

We revised figure 3 based on the suggestion from reviewer 1 to incorporate additional data points and add stationary-growth data. New representative images have also been incorporated in this figure.  However, we want to mention that all the original images were scanned at 300-dpi resolution for the reasons explained in our answer to Reviewer #1 comments. While we feel the current images are sufficient to visualize the roots and their curvature in response to clinorotation, we also recognize that they are not perfect. Unfortunately, this is the best we can offer at this time.

Figure 4

In my opinion the first panel of Figure 4A belongs to Supplementary material. The violin plot should remain. Please add the corresponding p-values to violin plot.

We agree with this suggestion and revised figure 4 accordingly.

Figure 5

5A. What does the arrow of g indicate in GS treatment?

5C. The table belongs to Supplementary material.

I strongly suggest to move Sup6A to main Figure 5.

The arrow with adjacent g indicates the direction gravity at the beginning of clinorotation or gravistimulation. We added the following sentence to the Figure legend to clarify this point (lines 451-453): “In panels (A) and (B), the direction of gravity(g)  at the onset of clinorotation (or gravistimulation: GS) is represented by a black arrow.“

We revised Figure 5 based on the reviewer’s suggestions.  We combined Figure 5A and Supplemental Figure S6A into a new Figure 5.  We also moved panels B and C from the original Figure 5 to supplemental Figure S6.   

Lns 392-395. Please discuss the biological significance of your finding

We added the following sentence to this paragraph to discuss the biological significance of this observation (lines 420-424): “These results are compatible with the initial few rounds of clinorotation leading to PIN relocalization toward a side of the columella cells dictated by the gravistimulus imposed during the first half-turn of clinorotation, lateral auxin transport and resulting curvature response.”

In addition to the changes discussed above, we also slightly modified a sentence in the Discussion (Lines 618- 623) to further clarify the point we were trying to make there: “While this approach was inspired from the presentation-time analysis carried out to evaluate gravisensitivity of plant organs, it is important to recognize that the Realization-Time parameter differs from the presentation time because in TR analysis the same stimulus is provided at each following inductive cycle during clinorotation, whereas a single limited gravistimulus is provided prior to horizontal clinorotation in presentation-time analysis.”

We would like to once again thank the reviewers for their excellent suggestions. We believe the changes we made to this manuscript in response to their feedback improved the quality of this document, and hope it is now ready for publication.

Round 2

Reviewer 2 Report

The authors have addressed a number of my concerns and I appreciate that. However, one significant issue I raised in my review was the quality of the Arabidopsis images used. Is the quality of the Arabidopsis images presented in the paper similar to the original uploaded files? The authors state in their response that “we also recognize that they (the images) are not perfect. Unfortunately, this is the best we can offer at this time”. 

I accept that scanning quality has impacted the image resolution and I have NO doubts about the soundness of the results presented in the article. However, in my opinion the quality of the images is still not adequate for publishing in IJMS. Based on this, it is impossible for me to approve this manuscript in its present form.

Author Response

Thank you for your feedback. In order to fulfill reviewer 2’s request, we used Photoshop to increase the representative images’ brightness and contrast without changing any shape and ratio of the images. Please see the revised figures (Fig. 1, 2, 3, 6 and supplemental Fig. S4).

We also added a sentence in material and method to clarify what has been done with the modification (lines 750-752) “To generate publication-quality pictures, we used Photoshop to increase the brightness and contrast of representative images, avoiding any shape distortion.

We appreciate very much the reviewer’s note about the soundness of the results presented in the article and hope the quality of the images has been sufficiently improved to warrant publication in IJMS.
